# *Candida parapsilosis sensu stricto* Antifungal Resistance Mechanisms and Associated Epidemiology

**DOI:** 10.3390/jof9080798

**Published:** 2023-07-28

**Authors:** Iacopo Franconi, Cosmeri Rizzato, Noemi Poma, Arianna Tavanti, Antonella Lupetti

**Affiliations:** 1Department of Translational Research on New Technologies in Medicine and Surgery, University of Pisa, Via San Zeno, 37, 56127 Pisa, Italy; iacopo.franconi@phd.unipi.it (I.F.); cosmeri.rizzato@unipi.it (C.R.); 2Department of Biology, University of Pisa, Via San Zeno, 37, 56127 Pisa, Italy; noemi.pomasajama@unipi.it (N.P.); arianna.tavanti@unipi.it (A.T.)

**Keywords:** *Candida parapsilosis*, antifungal resistance, antifungal tolerance, heteroresistance, epidemiology of antifungal resistance, antifungal susceptibility tests

## Abstract

Fungal diseases cause millions of deaths per year worldwide. Antifungal resistance has become a matter of great concern in public health. In recent years rates of non-*albicans* species have risen dramatically. *Candida parapsilosis* is now reported to be the second most frequent species causing candidemia in several countries in Europe, Latin America, South Africa and Asia. Rates of acquired azole resistance are reaching a worrisome threshold from multiple reports as in vitro susceptibility testing is now starting also to explore tolerance and heteroresistance to antifungal compounds. With this review, the authors seek to evaluate known antifungal resistance mechanisms and their worldwide distribution in *Candida* species infections with a specific focus on *C. parapsilosis*.

## 1. Introduction

*Candida* spp. infections have dramatically increased in the last twenty years [1]. Non-*albicans* species represent a rising concern in the hospital epidemiology of candidemia. Several reports define *Candida parapsilosis sensu stricto* as the second most frequent isolate from bloodstream infections, especially in Italy [2], Turkey [3,4] and Latin America—where in some reports, it was the first *Candida* species isolated from blood cultures [5,6]—Greece [7], South Africa [8] and Asia [9,10], also with worrisome azole-resistance rates.

*Candida parapsilosis sensu stricto* is a member of the commensal skin flora. Its role as a human opportunistic pathogen is seen mainly in immunocompromised subjects, low-weight at-birth newborns, onco-hematologic individuals and patients admitted to intensive care and burn units [11,12]. Invasive medical devices such as central lines and or other prostheses represent the main substrate of colonization and deep seeding due to its innate ability to form biofilm on organic or inorganic surfaces [13,14]. Plus, thanks to these intrinsic factors, *C. parapsilosis sensu stricto* is able to colonize inanimate materials and survive within the environment allowing for in-hospital spreading and patient-to-patient transmission via health workers’ hands as for multi-drug resistant bacteria [15].

Antimicrobial resistance is an issue in healthcare-associated invasive fungal infections [16,17] and despite the availability of new drugs [18] and the continuous research on alternative compounds [19,20,21,22], the increase in rates of antifungal resistance in most fungal infections and *C. parapsilosis*, in particular, is narrowing therapeutic options [23,24].

The aim of this review was to evaluate and depict a global picture of known antifungal resistance mechanisms in *Candida* species infections with a specific focus on *C. parapsilosis sensu stricto* and its worldwide epidemiology.

## 2. Antifungal Drugs and Associated Resistance Mechanisms in *Candida* Species in Comparison with *Candida parapsilosis*

### 2.1. Introduction to Antimicrobial Resistance in Medical Mycology

Antimicrobial resistance is the ability of a microorganism with no preexisting intrinsic resistance mechanism against anti-infective compounds to survive and even thrive in the presence of such antimicrobial drugs, delivered at recommended concentrations [16,17,25,26,27,28,29]. Antimicrobial resistance can be distinguished into intrinsic and acquired, independently from the pathogen, as it has been found in both fungi and bacteria. Intrinsic resistance is based on conformational aspects of the microorganism that are constitutionally present, such as molecular structures, enzymes and cellular components, targeted by selected drugs, that make it naturally non-susceptible to the selected agent. The European Committee on Antimicrobial Susceptibility Testing (EUCAST) defined it also as “Expected resistant phenotype” [30]. Intrinsic resistance is genetically determined in all the cells belonging to a single species and, therefore, present within the microorganism prior to exposure to the selected drug. In other terms, intrinsic resistance is inferred by: (i) the absence or (ii) the different conformation of the target of a specific drug, making it not feasible for the antimicrobial agent to exert its activity [31,32,33]. Acquired resistance, on the contrary, represents the development of a resistant phenotype of a specific microorganism, which was not known to harbor any intrinsic resistance to the selected drug, and it is associated with prolonged drug exposure [16,18,26,34,35].

Parallel to these definitions, the authors have also identified two distinct scenarios for antimicrobial resistance: (i) microbiologic resistance, known as the ability of the microorganism to grow when exposed to adequate concentrations of an anti-infective drug normally active on wild-type strains and (ii) clinical resistance where the antimicrobial compound is unable to eradicate the in vivo infection occurring in a patient despite the demonstrated in vitro susceptibility of the infectious strain to the selected and currently administered antimicrobial therapy [16,17,26,28,36,37].

Therefore, when referring to antimicrobial resistance as a whole, it can be described as the lack of microbial growth inhibition usually achieved by the effective antimicrobial compound administered at regular dosage reaching adequate in vivo concentrations. This lack of growth inhibition can be directly observed in in vitro testing with Minimum Inhibitory Concentrations (MICs) for the selected drug falling outside the susceptible interval range. It can also be drawn by clinical worsening of the patient with persistent isolation of the pathogen from the same clinical samples despite the administration of the anti-infective drug already proven to be effective at the in vitro susceptibility tests [16,17,26,36,37]. Finally, the two main driving factors for the development of both clinical resistance and persistent isolation of the pathogen from patients’ clinical samples are (i) the ability of *Candida* spp. to form biofilm and (ii) reduced drug concentrations achieved at the infected site.

When evaluating antimicrobial resistance in fungi, especially for yeast infections of the *Candida* genus, the Clinical and Laboratory Standards Institute (CLSI) and EUCAST provided, over the years, two broth microdilution standardized methods of in vitro susceptibility testing that has been accepted by the scientific community as the two reference techniques to evaluate antifungal susceptibility, allowing for detection of resistant phenotypes for several *Candida* species [26,37,38,39].

Various genetic and molecular mechanisms underlying antifungal resistance have been extensively studied in *Candida albicans*, as this is the most frequently isolated *Candida* species implied in human infections [17,40]. Gene editing and CRISPR-Cas9-based techniques have been the primary tool to confirm the role and the effects of newly found mutations in *Candida* species [41]. However, the complexity of antifungal resistance resides in the co-existence within a single resistant strain of several mechanisms, each of which independently contributes to the non-susceptible phenotype. Therefore, acknowledging the main driving mutation or resistance mechanisms might be hard to accomplish [16,17,36]. Despite advances in research, several antifungal resistance mechanisms in different species need to be further evaluated in consideration of the fact that in recent years non-*albicans* species have become a real matter of concern for public health [16,17,26,27].

### 2.2. Azole Resistance

Azole drugs are major antifungal compounds that have been extensively used in clinical practice. Their mechanism of action is to bind and inhibit the lanosterol 14-alpha-demethylase [42,43]. This enzyme is present within several fungi and its activity is associated with ergosterol synthesis, a fungal cell membrane component [44,45]. Azole compounds, especially fluconazole, used to be the drug of choice prior to the advent of echinocandins and the first report of suggestion of echinocandin as the first line of therapy in candidemia patients is stated in the ESCMID 2012 guidelines [46]. The same assumptions and recommendations were adopted by the Clinical Practice Guidelines for the Management of candidiasis by the Infectious Diseases Society of America in 2016 [47]. Generally speaking, *Candida* species have developed three distinct molecular mechanisms through which they exert azole resistance. The most frequently encountered in clinical practice is associated with the increased activity of efflux pumps [48,49,50], which are fungal membrane proteins that fall in the molecular domain of ATP-binding cassette (ABC) transporters and major facilitator superfamilies (MFS) that can be found also in bacteria, plants and animals [51,52]. Their molecular activity aims at removing the drug from within the microorganism. The binding of the efflux pump to the azole compounds leads to the excretion of the antifungal molecule and, therefore, plays a pivotal role in the development of drug resistance [43,44,50,53]. The molecular ways through which overexpression of the efflux pump is achieved are represented by specific mutations in the transcription factors genes as gain-of-function mutations [50,54,55]. The type of efflux pump overexpressed may vary upon the *Candida* species exposed to azole compounds as several different transporters have been reported to be overexpressed according to different *Candida* species, despite similar underlying genetic mechanisms. For example, Cdr1p and Cdr2p are two transmembrane transporters that belong to the ABC-Transporter superfamily that have been reported to be overexpressed in azole-resistant *Candida albicans* strains [27,53]. Within the same superfamily, CgCdr1p, CgPdh1p and CgSnq2p are efflux pumps present in *Candida glabrata*; CkAbc1p and CkAbc2p in *Candida krusei* and Cdr1p in *Candida auris*. Among efflux-pumps that belong to the other superfamily (MFS-Transporter), Mdr1p has been found to be overexpressed in *Candida albicans* while CgQdr2p and CgFlr1p in *Candida glabrata* [27,49,53,56,57,58].

In the case of *C. albicans* the three major efflux pumps mentioned above detected in azole-resistant clinical isolates are encoded by corresponding genes *CaCDR1*, *CaCDR2* and *CaMDR1*. The transcription factor CaTac1p regulates the expression of the first two efflux pump genes, while CaMrr1p, another transcription factor, controls the expression of the last one mentioned. In their corresponding genes, *CaTAC*1 and *MRR*1 authors found several gain-of-function mutations leading to increased levels of expression of the three related efflux pumps [59,60,61,62].

The same molecular mechanisms with different transcriptional genes involved have been described in *C. glabrata*, for which, gain-of-functions mutations in the gene *CgPDR1* encoding for the transcription factor of ABC-T pumps *CgCDR1*, *CgSNQ22* and *CgPDH1* correlates with increased expression of corresponding efflux pumps resulting in azole resistance [63].

The second and third resistance mechanisms described for azole resistance are related to mutations in the genes directly encoding enzymes correlated to ergosterol synthesis or their transcription factors. The second mechanism is known as “target-mutation”, while the third leads to ergosterol overexpression. To this point, the most frequently reported gene hosting in vivo mutations in azole-resistant *Candida albicans* and *Candida glabrata* strains is *ERG11* [56,63,64]. *ERG11* encodes for a cytochrome P450 known as Erg11p [65], which has a sterol-14α-demethylase activity converting lanosterol into 4,4-Dimethylcholesta-8,14,24-trienol [65]. Azole compounds, in particular fluconazole, can bind and disrupt this fungal metabolic pathway. They act specifically on this enzyme, leading to the accumulation of a toxic metabolite, 14 alpha-methyl-ergosta-8,24(28)-dien-3 beta,6 alpha-diol. This toxic metabolite causes yeast cell death through its intracellular accumulation [42,65]. As stated above, *Candida* spp. have developed two distinct ways to overcome this detrimental effect: (i) overexpression of the target gene and (ii) gene target mutations, altering the binding site where usually the effect of the antifungal drug is elicited [27,44,66,67,68,69,70,71]. To the first point, authors have highlighted the presence of gain-of-function mutations in the transcription factor genes implied in the regulation of *ERG11* expression. Such genes are known as *UCP2* and *NTD80* and have been found in both *C. albicans* and *C. glabrata* azole-resistant strains [72,73,74]. Gain-of-function mutations associated with an increased expression of the *UPC2* gene are A643V, G648D, G648S and Y642F [73].

On the second mechanism, several point mutations have been linked to actual in vivo and in vitro azole resistance due to target mutations. For example in *C. albicans*, Xiang et al. [70] reported five different point mutations (Single Nucleotide Polymorphisms, SNP) in *ERG11* correlated with azole resistance and investigated their structural position on a 3D model of the target enzyme. Their work demonstrated that amino acid substitutions caused by such mutations were all located near the substrate channel of the target enzyme (A114S, Y132F, Y132H, K143Q and K143R) or the active binding site (G472R) [70]. Other point mutations with the same effect on azole resistance have emerged from the study conducted by Silva and colleagues regarding *C. glabrata* (C108G, C423T and A1581G) and *C. krusei* (Y166S, G524R) [75]. Other genes connected to the ergosterol synthesis which may play a role in the development of resistance are *ERG2*, *ERG3* and *ERG6*. Such genes encode for enzymes that convert the intermediate product of the ergosterol synthesis after exposure to azole and inhibition of the Erg11 cytochrome P450 generating toxic metabolites that compromise cell growth and vitality in both *C. albicans* and *C. glabrata* [76,77,78,79,80]. Mutations targeting these genes, causing gene disruption, led to the acquisition of an azole-resistant phenotype at in vitro studies since the conversion of the intermediate metabolite into the toxic one was blocked and yeast pathogen could withstand the azole-induced inhibition of the Lanosterol-14α-demethylase [78,79,81].

Gain-of-function and point mutations are not the only genetic mechanisms underlying an azole-resistant phenotype. Aneuploidy, altered mismatch repair, loss of heterozygosity, increase in number of copies of target genes and trisomy of selected chromosomes that incorporate ABC-Transporters, MFS or *ERG11* genes, have all been demonstrated to elicit a resistant phenotype in a previously susceptible one [26,27,50,82,83,84,85].

It is still a matter of debate whether alterations in the azole-intake pathway could play a role in inducing azole resistance or not. The contribution of azole import to resistance has yet to be elucidated since the actual protein implied in the transmembrane transportation carrying the drug into the yeast cell has not been described so far. Despite this, the kinetics of azole accumulation into *Candida* spp. do not reflect those of passive diffusion and, therefore, the role and the presence of a possible carrier-protein have been postulated [27,50].

Another major antifungal drug within the azole family is isavuconazole. Such compound is relatively new and has shown promising in vitro activity against the most frequently encountered *Candida* species in clinical practice, such as *C. albicans*, *C. glabrata*, *C. parapsilosis*, *C tropicalis*, *C. krusei*, *C. kefyr* and *C. lusitaniae* [86,87]. Moreover, isavuconazole is currently referred to as an alternative treatment of invasive aspergillosis, a therapeutic option for mucormycosis and a potential oral-step down therapy in the treatment of candidemia, according to the ACTIVE trial results [88,89]. However, despite its relatively short period of clinical use, isavuconazole resistance has been described [89,90]. Main mechanisms of isavuconazole resistance were found in azole-resistant *Candida* species overexpressing the *CDR* genes, same results were not observed in those *Candida* spp. with increased expression of *MDR1* gene [91]. In addition, also *ERG11* and/or *ERG3* mutations were found to be associated with the development of isavuconazole resistance [90,91]. Azole resistance in *Candida parapsilosis sensu stricto* has become a clinically relevant issue in the last decade [92], with the World Health Organization introducing this opportunistic pathogen among the high-priority group of yeast and fungal infections [23]. Major mutations linked to clinically demonstrated acquired resistance to azole compounds and associated affected molecules are shown in Table 1.

*C. parapsilosis sensu stricto* had no intrinsic resistance to azole drugs, therefore, reports and studies on azole-resistant hospital outbreaks are related to acquired resistance [11]. In an early study conducted by Silva and colleagues, three *C. parapsilosis* azole-susceptible strains were exposed to fluconazole, voriconazole or posaconazole in order to induce resistance, then gene expression of *ERG11* and efflux pumps were analyzed [93]. The results demonstrated that the resistance mechanism could be associated with G583R and K873N amino acid substitution mutations in the transcription gene of a MFS *MDR1*, known as *MRR1* for the fluconazole- and voriconazole-resistant strains. In the same study upregulation of *UPC2* and *NDT80* genes encoding for transcriptional factors increased the expression of *ERG11* resulting in posaconazole resistance [93]. To this point, Arastehfar et al. identified two amino acid substitutions P45H and Q371H in the *UPC2*, leading to its overexpression, in fluconazole-resistant and voriconazole-susceptible-to-intermediate strains of *C. parapsilosis* [4].

In another study still conducted by Silva and colleagues [109], induction of azole resistance was obtained after exposing the yeast pathogens to several antifungals at different gradients. Their results pointed out that among azole compounds, fluconazole exposure took 15 days to induce resistance in previously susceptible isolates whether along the same period, no change in the susceptibility patterns of posaconazole was observed [109]. Plus, authors reported that induced fluconazole resistance would also affect susceptibility to voriconazole and vice versa; however, these two compounds showed no induced cross-resistance to posaconazole [109]. Surprisingly, after elimination of the azole pressure and subsequent cultures without further exposure to the previously mentioned antifungal compounds no substantial change was observed in the resistant susceptibility profiles. Plus, the same strains underwent treatment with known efflux pump inhibitors. Incredibly, these isolates did not revert the acquired resistant phenotype presenting high MIC values for fluconazole and voriconazole. This brought to the conclusion that the eventual underlying resistant mechanism could not be referred to efflux pumps since both the removal of the drug and the treatment with efflux pump inhibitors had no impact on the acquired resistance mechanism [109].

The residual susceptibility to posaconazole in fluconazole- and voriconazole-resistant *C. parapsilosis* strains can be explained partially because of the number of domains in the target site of the lanosterol-14α-demethylase that are bound by the different azoles. For instance, both fluconazole and voriconazole present only one binding site while posaconazolee has two of them. This is also the reason why overexpression of *ERG11* is the resistance mechanism for posaconazole [93]. To this point, it is important to mention that also upregulation of *MDR1* does not affect posaconazole susceptibility since this compound is a poor substrate of the previously mentioned efflux pump [53,110].

One more aspect of azole resistance should be further elucidated, as the two transcription factor genes *UPC2* and *NTD80* implied in the expression of enzymes correlated with ergosterol synthesis (*ERG11 ERG2 ERG3 ERG4 ERG6 ERG25*) reported in *C. albicans* were also found to be overexpressed in fluconazole-, voriconazole- and posaconazole-resistant *C. parapsilosis* [98]. In fact, their deletion restored complete susceptibility to all these antifungal compounds, however, in the same study, Branco et al. found that disruption of *UPC*2 had a more incisive reduction in MIC values of azole drugs than *NTD80* [98].

Interesting findings on the underlying azole-resistance mechanism resulted from an experiment conducted by Souza and colleagues [95] on nine strains of fluconazole-resistant *C. parapsilosis*. Mutations in the *ERG11* and in the efflux pump genes were explored. The results showed that all resistant strains harbored a missense mutation in the *ERG11* gene generating the following amino acid substitution Y132F. This kind of mutation changed the protein structure leading to loss of binding activity with fluconazole [95], still other resistance mechanisms were found as overexpression of *ERG11*, *CDR1* and less frequently *MDR1* [95].

In a study on a Brazilian ICU cohort of COVID-19 patients with candidemia due to fluconazole-resistant *C. parapsilosis*, Daneshnia et al. found that only 35.1% of isolates showed the K143R mutation in the *ERG11* gene. Interestingly, all fluconazole resistant isolates presented the L518F mutation in the *TAC1* gene, which is a transcription factor of *CDR1*, which was demonstrated to be a causative mutation of acquired fluconazole- and voriconazole-resistance in the same study [100]. Berkow and colleagues identified two more mutations in the *TAC1* transcription factor gene of *C. parapsilosis* (G650E and L978W) correlated with overexpression and upregulation of the target efflux pump Cdr1p with acquired fluconazole- and voriconazole-resistant phenotypes [97].

There is a well-established relationship between the type of efflux pump overexpressed and the associated resistance spectrum for *Candida* species other than *C. parapsilosis*. Overexpression of *CDR* efflux pump class, but not *MDR,* shows cross-resistance to all antifungal azole drugs, while the second class only affects mainly fluconazole [36]. However, regarding *C. parapsilosis*, Branco et al. [103] reported a case of cross-resistance between fluconazole and voriconazole directly correlated with a specific mutation G604R that induced overexpression of the *MRR1* transcription factor gene resulting with the overexpression of the Mdr1 efflux pump [103].

Finally, a study conducted by Grossman et al. provided a great effort in elucidating the most frequent resistance mechanism for azole resistance in *C. parapsilosis* [111]. In their study, these authors evaluated and sequenced the genome of 30 fluconazole-resistant isolates obtained from blood-stream infections, demonstrating that 57% presented a SNP in the *ERG11* gene resulting with the Y132F amino acid substitution previously mentioned [111]. Anyhow, also overexpression of *MDR1* was registered, however, its frequency was less observed than the previous mutation. These authors reported that SNP correlated with *MRR1* were more difficult to investigate and further research would have been required [111]. To this point, Branco and colleagues later on described two missense mutations into the *MRR1* gene with amino acid substitution G583R and K873N imputable of determining the fluconazole- and voriconazole-resistant phenotype [94]. Still, data from a recent world-wide surveillance study conducted by Castanheira et al. analyzing multiple strains from different countries clearly determined that azole-resistance in *C. parapsilosis* was mainly driven by Y132F substitution in the *ERG11* gene with a smaller role played by efflux pumps [112].

Last, it is important to mention that in a study from Arasthefar et al. [4] conducted during a clonal outbreak of candidemia due to azole-resistant *C. parapsilosis*, in addition to the Y132F substitution in *ERG11*, also the substitution K143R was described. Such amino acid change had been previously found also in *C. albicans*. Still, another important gene should be mentioned, as the *ERG3* gene, which is also implied in the ergosterol synthesis, has been found to be target of point mutations with consequent development of azole resistance in *C. parapsilosis*. To this point, Branco et al. found a specific missense mutation (R135I) that led to loss of function of the enzyme in a posaconazole-resistant isolate [98].

Data regarding species-specific resistance mechanisms to isavuconazole in *C. parapsilosis* are scarce, however the previously mentioned mechanisms, described for other species, as overexpression of *CDR1* gene and *ERG11* target mutations, could be also found in *C. parapsilosis*. Anyhow, reports highlight that only a relatively small proportion of *C. parapsilosis sensu stricto* are non-wild-type and/or resistant to isavuconazole, as reported by Desnos-Ollivier et al. [87] and Marcos-Zambrano et al. [86], being respectively 0.8 and 1.1%.

Within the psilosis complex, in the context of azole-resistance, authors have reported a specific mutation in *C. orthopsilosis* known as A395T mutation in the *CoERG11* gene. Such mutation is associated with a non-synonymous amino acid substitution Y132F and was proven to induce azole-resistance in previously susceptible *C. orthopsilosis* isolates [113]. Data regarding azole-resistance mechanisms in the two members of the psilosis group other than *C. parapsilosis* are anyhow lacking, however, it might be helpful to highlight that reports from different countries at in vitro tests showed very low rates of non-wild type MIC phenotypes for both *C. orthopsilosis* and *C. metapsilosis* for azole compounds [114,115].

An overview of major azole-resistance mechanisms for *C. parapsilosis sensu stricto* is depicted in Figure 1.

Five major mechanisms connected to azole-resistance in *C. parapsilosis*. SNPs in the *TAC1* and *MRR1* gene are associated with their overexpression and consequent upregulation of their targets, Cdr1p and Mdr1p respectively. SNPs in the *UPC*2 and *NTD80* genes are associated with overexpression of *ERG11*, SNPs in the *ERG11* gene alter the yeast target enzyme of azole compounds. SNPs in the *ERG3* gene inducing loss of function mutations reduce the conversion of intermediate azole compounds in toxic metabolites with increase of Ergosta-7-enol (yellow oval) that could replace ergosterol (blue oval) in the fungal cell membrane without altering its molecular structure stability. (Image Created with BioRender.com).

### 2.3. Echinocandin Resistance

Echinocandins are a group of antifungal drugs that target specifically the β-(1,3) D-glucan synthase, which is encoded by two genes *FKS1* and *FKS2* that to a certain extent are redundant [116]. Glucans are polysaccharide components of the fungal cell wall, and their synthesis inhibition by echinocandins leads to cell death [116]. Precisely, the non-competitive molecular bond is established by the drug and a specific subunit of the fungal enzyme, known as Fks1p [117]. Their spectrum is broader than fluconazole and rates of fungal eradication were reported to be higher than fluconazole [118]; therefore, echinocandins are the recommended treatment in case of candidemia and invasive fungal infections due to *Candida* spp. as first-line empiric therapy [47].

Resistance to echinocandins is reported to be below 1% in *C. albicans* [40] clinical isolates and less than 10% in *C. glabrata* [119] and it has been linked to point mutations in the genes that encode the β-(1,3) D-glucan synthase causing an amino acidic substitution in the active-binding site of the target enzyme. A 645 serine to proline (S645P), phenylalanine (S645F) and tyrosine (S645Y) substitution in the Fks1p subunit triggers the development of echinocandin-resistance in *C. albicans* [28,120,121,122,123]. S645P substitution was reported to be the most prevalent among *C. albicans* [123]. Similar mutations have emerged in *C. glabrata* and *C. krusei* [124]. Caspofungin, micafungin and anidulafungin are all affected by these mutations, both in hetero or homozygosis since they are dominant and always associated with elevated MIC values [104]. In *C. glabrata* specifically, mutations in the *FKS2* gene have been associated with a major impact on resistance than those present in the *FKS1* gene [125]. Echinocandin-resistant phenotypes usually correlate with non-susceptibility to all antifungal drugs of the class with the exception of a single mutation in the *FKS2* gene (Fks2p-S663F). This mutation was found in a *C. glabrata* strain where authors described a loss of drug activity for anidulafungin and caspofungin but not for micafungin [126]. Anyhow mutations obtained from clinical isolates in the *FKS1-2* hot spot regions are known to affect the entire class of antifungal drugs [33,126].

Mutations in the hot spot region of *FKS1* and *FKS2* are not the only resistance mechanism described in echinocandin-resistant *C. albicans* and *C. glabrata* [127]. To this point, response to stress conditions may play a pivotal role, especially when fungal pathogens are exposed to echinocandins with alteration of the cell wall. In fact, when the integrity of the cell wall is disrupted, due to the β-(1,3) D-glucan synthase inhibition, studies have demonstrated that Rho1, a GTP-ase protein that represents the second subunit of the β-(1,3) D-glucan synthase gets activated. In the Fks1p subunit, which is the other subunit of the β-(1,3) D-glucan synthase, as previously mentioned, resides the catalytic activity of the enzyme, the actual site where the β-(1-3) D glucan is synthesized, that is targeted by echinocandins, while in the other subunit, Rho1, resides the regulatory activity of the enzyme itself. Authors speculate that activation of Rho1 may induce overexpression of the β-(1,3) D-glucan synthase while also triggering intracellular signaling of the protein kinase C (PKC) enabling fungal cell to activate a series of stress-responses, to compensate and restore the integrity of the cell wall through increase in chitin synthesis [128,129,130]. Also Ca^2+^/calcineurin, an intracellular stress-response pathway, contributes to the increase in chitin synthesis [131]. Last, but not least, Rho1 does not only activate PKC intracellular signaling pathway but it also provides upregulation of the *FKS* genes [127]. Despite being extensively studied in vitro, the clinical relevance of the above-mentioned molecular mechanisms and their role in treatment failure and antimicrobial resistance have yet to be demonstrated.

As for *C. parapsilosis sensu stricto* echinocandin-resistance has been less frequently reported than azole-resistance. Echinocandins exert a fungistatic effect on *C. parapsilosis* differently from the fungicidal activity displayed on other *Candida* species. This is due to a constitutional amino acid change in one of the hot spot regions of the Fks1p found to be naturally present in this kind of fungal species [104]. All mutations and relative effects on echinocandin susceptibility profile in *C. parapsilosis* are reported in Table 1. This constitutional substitution reported for *C. parapsilosis* accounted for its intrinsic reduced susceptibility to echinocandins associated with MIC values higher than other *Candida* species. Such intrinsic mutation was found in the hot spot region 1 of the subunit Fks1p and it was a proline to alanine substitution (P660A) [104]. This naturally occurring polymorphism has been also detected within the other species of the *psilosis* group like *C. orthopsilosis* and *C. metapsilosis* [104], still non-wild type phenotypes for such species for echinocandins are rarely seen in clinical practice [132,133,134], but the rarity of the isolation of such species does not allow to draw firm conclusions as more studies are needed to evaluate prevalence of non-wild type phenotypes for echinocandins. However other mutations previously described in echinocandin-resistant *C. albicans* or *C. glabrata* strains that were found within the hot spot region of the *FKS1* and *FKS2* genes were not present in echinocandin-resistant *C. parapsilosis* in the study of Martì-Carrizosa [105]. Indeed, they found that both mutations V595I and F1386S detected in echinocandin-resistant *C. parapsilosis* isolates were placed outside the hot spot regions [105]. These mutations were previously reported by Johnson and colleagues to be associated with acquired resistance to echinocandin in *Saccharomyces cerevisiae* [135]. Similar findings were reported by one study from a Brazilian outbreak of fluconazole-resistant and echinocandin-tolerant *C. parapsilosis* causing candidemia among COVID-19 patients. In this study, the authors linked the previously cited mutations to echinocandin-tolerant phenotype, and highlighted the presence of another specific mutation E1393G in the *FKS1* gene, which was also linked to echinocandin-tolerance [100]. Further investigations correlated the presence of such mutation along with the above mentioned V595I, S745L and F1386S with the in vitro development of echinocandin-resistance [107]. Still, other studies reported also hot spot regions of the Fks1p to be the target of specific mutations affecting negatively echinocandin-susceptibility. For example, a recent study found that R658G mutation in the hot-spot region 1 of Fks1p was associated with a micafungin-resistant phenotype. This substitution was discovered in four micafungin-resistant *C. parapsilosis* strains isolated from blood cultures in 2020 [106]. A report from a Chinese study of a *C. parapsilosis* pan-echinocandin-resistant strain isolated from blood cultures revealed the presence of another mutation known as S656P still in the hot spot region 1 of Fks1p [108].

Surprisingly, another mechanism of echinocandin resistance was reported by Ryback et al. observing that a mutation G111R in *ERG3* correlated with an increase in all echinocandins MIC. This was the first study to ever correlate an *ERG3* loss of function with an acquired resistant phenotype to echinocandins, even though identification of the actual resistance mechanism is still matter of research [99].

Worthy of mention, especially in the case of *C. parapsilosis senso strictu* is rezafungin, which is a second generation echinocandin. Its pharmacokinetic/pharmacodynamic properties allow for a reduction in liver toxicity with a prolonged half-life, exerting the same inhibition observed for all other echinocandins on the β-1,3-D-glucan synthase [136]. By looking at the distribution of MIC reported within the psilosis group, *C. parapsilosis sensu stricto* showed higher MIC values (4 μg/mL) for this molecule, while *C. metapsilosis* was 0.5 μg/mL and *C. orthopsilosis* was 1 μg/mL [137]. As expected, all three psilosis species demonstrated MIC values higher than those reported for all other *Candida* species, due to their previously mentioned natural polymorphism [138]. Up to 2021, no resistance to rezafungin in *C. parapsilosis sensu stricto* was documented [139]; however, in 2022 Siopi et al. reported a case of pan-echinocandin *C. parapsilosis sensu stricto*—including also rezafungin—with an isolate harboring a new mutation in the HS region of the *FKS1* gene (F652S) [140].

### 2.4. Polyene Resistance

Polyenes are a class of drugs that comprise Amphotericin B (AMB), Nystatin and Amphotericin A, with the first recognized as a major systemic antifungal drug and one of the first to be used in clinical practice [141]. AMB mechanism of action resides in the ability of the molecule to bind the ergosterol in the fungal membrane resulting in pore formation and loss of intracellular electrolytes causing lastly cell death [142]. AMB showed a broad spectrum of activity exerting a fungicidal effect on several *Candida* spp. and filamentous fungi [143,144]. Despite the long time since its introduction in clinical practice, rates of acquired resistance to AMB remained low and only rare cases have been reported [33,145]. Among yeast pathogens *C. glabrata*, *C. krusei*, *Candida haemulonii*, *C. lusitaniae*, *C. auris* and *C. guillermondii* are species that have been most frequently associated with AMB-resistance [126,145]. *C. parapsilosis* was listed among the AMB-susceptible fungal isolates [146].

Authors suggest that a reduction in the ergosterol composition of the fungal membrane associated with *ERG2*, *ERG3* and *ERG6* loss of function mutations might represent the underlying AMB-resistance mechanisms [33,68,126,145,147,148], since they reduce the amount of ergosterol present in the fungal membrane and, therefore, the target of AMB itself [149,150]. In addition, also up-regulation of *ERG5*, *ERG6* and *ERG25* has been associated with acquired AMB-resistance, since this modification led to the synthesis of a different sterol than ergosterol. Once inside the fungal cell membrane this new sterol intermediate displays a reduced binding activity to the antifungal drug ensuring anyhow structural stability to the yeast [78,145]. Still, upregulation of *ERG* genes and loss of function mutations are not the only mechanisms responsible for AMB-resistance. As observed for echinocandins, the stress-response may play an important role in the survival of the fungal cell. AMB-induced membrane alteration and consequent oxidative stress induces the acquisition of a resistant phenotype by increasing composition in chitin content of cell wall and by reducing fluidity of the membrane [151]. To this point, authors have highlighted another fungal resistance mechanism activated after exposure to AMB, which is adaptation and response to drug-induced oxidative stress. Fungal pathogens under AMB drug pressure might develop an increase in levels of oxidative stress-response proteins, such as catalase and heat shock protein 90 (*HSP90*), countering the negative effects of reactive oxygens species [130]. A latter mechanism of resistance reported by Healey et al. [152] in 2016, that is not exclusively related to AMB, focused on genes related to mismatch repair in *C. glabrata*. Disruption of the *MSH2* gene increased mutation rates among other genes normally involved in resistance to azoles, echinocandins and also AMB leading to a multi-drug resistant phenotype [152].

Despite these findings, AMB-resistance mechanisms in *C. parapsilosis sensu stricto* need to be further investigated and thankfully it is still a rare phenomenon with reports showing an extremely low rate of resistance among different countries and across different regions. [153].

### 2.5. Flucytosine Resistance

Protein and DNA synthesis are the metabolic target pathways of flucytosine [154]. After administration, flucytosine gets transported into the yeast cell thanks to a cytosine permease and converted to 5-Fluorouracile (5-FU) by a cytosine deaminase present within the yeast pathogens. Next 5-FU gets converted to 5-fluorouridine triphosphate and 5-fluorouridine monophosphate, the first compound alters protein synthesis interfering directly with amino-acylation of tRNA once it has been integrated in the RNA molecule [155,156]. On the other hand, thymidylate synthase is the target of the second active metabolite of 5-FU, resulting in inhibition of DNA synthesis [157]. The resistance mechanisms described for this drug were loss of function mutations correlated with the genes encoding proteins implied in the import of the drug as cytosine permease (*FCY2*), or involved in its intracellular metabolism as cytosine deaminase (*FCY1*) and uridine monophosphate phosphorylase (*FUR1*) [158,159,160]. Such mutations lead to a reduction of these target enzymes reducing both the uptake and the metabolism of flucytosine. Another described mechanism for flucytosine resistance in some resistant *Candida* spp. strains appears to be overexpression of the substrate increasing pyrimidine synthesis [161,162]. It is also important to mention that *Candida* species develop flucytosine-resistance rapidly after treatment exposure, even within 48 h after initiation, therefore, international guidelines do not recommend monotherapy with flucytosine supporting instead combination therapy with AMB or azole compounds in selected cases [47].

Data on flucytosine-resistance in *C. parapsilosis sensu stricto* are scarce, since its use in clinical practice as a single drug agent in monotherapy is commonly avoided. The first report of development of flucytosine-resistance during therapy in *C. parapsilosis* was described by Hoeperich et al. in 1974 [163], where authors found a reduced cytosine deaminase activity in the resistant strain. No further molecular investigations could be performed at the time to evaluate underlying resistant mutations. However, along with the previously mentioned resistance mechanisms, Sun et al. suggested an adjunctive genetic adaptation in *C. parapsilosis* regarding yeast response to flucytosine [164]. *C. parapsilosis* showed to have ortholog genes encoding for the same enzymes implied in the metabolism of flucytosine. In addition, this species was found to have chromosome aneuploidy, in particular trisomy of the chromosome 5 as a potential response and adaptation to the drug [164]. However, such genetic modification does not fully evolve in clinical resistance, anyhow Sun et al. reported that it was correlated with increased antifungal tolerance [164]. To this point, it is worthy to mention that primary objective of their study was to investigate the effect of such genetic modification on caspofungin-resistance and/or tolerance. Later, the authors found a cross-adaptation to flucytosine as they observed an increased tolerance [164]. The biological explanation and interpretation given by the authors refers to a particular gene that is normally found on chromosome 5 in *C. parapsilosis* that encodes for chitin known as *CHS*7. Such gene ends up to be overexpressed under trisomy conditions like in this case [164,165].

### 2.6. Antifungal Tolerance

The concept of drug tolerance was first introduced when observing bacterial isolates able to survive in the presence of antibiotics at concentrations above the MIC without any known underlying resistance mechanism [166,167]. This phenomenon was reflected in vitro by a slow growth of a small proportion of cells within a single colony of the microorganism, showing tolerance to the specific antimicrobial drug [16,29,166]. Such microorganisms, however, did not harbor any known resistance mechanism and once tested again for the selected antimicrobial molecule only a small sub-proportion of them still grew under MIC concentrations, suggesting that tolerance should be referred to a peculiar physiological and/or epigenetic state of the microorganism instead of genetic acquisition of a resistant phenotype [166]. Therefore, at in vitro antifungal susceptibility tests tolerant isolates are included in the susceptible category and cannot be distinguished by non-tolerant ones due to their slow growth [26]. In vitro demonstration of this phenomenon has been defined as “trailing growth” at broth microdilution methods, where wells in which antimicrobial drugs were present at an inhibitory concentration hosted a slow growth of the pathogen [166]. The same concept can be translated in yeast pathogens especially in *Candida* species [16,168,169]. Authors impute tolerance to the presence of persister cells among the microorganism population tested for antifungal resistance [170,171], as others suggest that aneuploidy might also be involved [172]. As stated by Berman and colleagues in regards of fungal microorganisms, tolerance is the ability of yeasts to slowly grow above MIC values [26]. Such growth would not be detectable before 48 h of incubation [26]. The same authors proposed and hypothesized that different cellular stress-responses among fungi of the same isogenic population might be an explanatory factor contributing to drug tolerance, but data confirming such assumption have yet to be provided [26]. In *C. albicans* studies pointed out that a different composition in the sphingolipid profile in cell membrane might be involved in the development of azole-tolerance [173]. Also increased chitin synthesis—especially with echinocandin molecules—may play a significant role in defining antifungal tolerance favoring survival of the yeast cells and slow growth rate after 24 h incubation [131,174]. Clinical consequences of drug tolerance represent a fervid field of research, with several authors reporting a correlation with the development of antimicrobial resistance [107,127,175,176] and with mortality and therapeutic failure even in fungal diseases [176,177]. Antifungal drug tolerance varies from one class to the other as it has been more frequently observed with azole compounds rather than echinocandins, since the first class mentioned is known to have a fungistatic effect [16,92]. Indeed, the proportions of *C. parapsilosis sensu stricto* and *C. glabrata* cells found within the in vitro trailing growth, that are able to grow slowly under drug concentrations higher than the MIC values, are different between azole and echinocandins [92,127], with more than 1% of total fungal population tolerant to azole drugs and less than 1% to echinocandins, although this last class of drugs has only a fungistatic effect on *C. parapsilosis* [92]. Antifungal tolerance is difficult to assess via routinely available in vitro tests, therefore, authors proposed specific tests to achieve valuable and interpretable results, such as for echinocandin tolerance in the study from Daneshnia [100]. In this study, *C. parapsilosis* cells were incubated in RPMI1640 liquid medium added with the intermediate breakpoint micafungin value (4 μg/mL) according to CLSI, and plating was performed at several time intervals comparing Colony Forming Units (CFU) with untreated controls [100]. In this study, the authors reported several mutations implied with echinocandin-tolerance in *C. parapsilosis* in addition to those previously reported as S745L (found outside the Hot Spot region 1 of the *FKS1* gene) and A1422G and M1328I (found both outside the Hot Spot region 2 in the *FKS1* gene) [107]. Another in vitro test proposed by Berman and colleagues is the “fraction of growth” [26]. This test compares fungal growth within the MIC inhibition zone on solid medium of tolerant colonies after prolonged incubation time (48 h) and the fungal growth observed beyond the same area. Such measurement allows also for an estimation of the degree of tolerance. [26]. In order to do so, the authors also rely on the use of automated software to estimate such distance quantifying the grade of antifungal tolerance [26]. Even liquid medium tests have been proposed to evaluate such microbiological phenomenon, falling under the name of “Supra MIC growth” [26].

To better elucidate the relevance of antifungal tolerance and its implications on the development of echinocandin resistance, it is mandatory to mention a study from Daneshina et al. [107] where the in vitro selection of echinocandin-resistant *C. parapsilosis* isolates happened only in echinocandin tolerant cells after being exposed and plated on agar solid medium supplemented with echinocandin highlighting an inducible resistant phenotype from tolerant yeast strains [107].

### 2.7. Heteroresistance

Heteroresistance was firstly described in bacterial microorganisms as *Staphylococcus* spp., *Acinetobacter* spp., *Myocobacterium tuberculosis* and then in a fungal opportunistic pathogen, *Cryptococcus neoformans* [178,179,180,181]. As for tolerance, it is a microbiological phenomenon that takes place in a very reduced subset of microorganisms within a bacterial or fungal population differing from the previous one as it happens in one cell in 10^5^–10^6^ CFU of susceptible colonies. Although rarer than tolerance, it correlates with a detectable resistant phenotype at in vitro [167]. Fungal pathogens showing heteroresistance may reach up to eight-fold the MIC values registered in common in vitro susceptibility tests; however, genetic resistance and heteroresitance remain two distinct microbiological phenomena [26]. For example, in two yeast pathogens, *C. glabrata* and *C. neoformans*, heteroresistance to fluconazole was observed in less than 1% of fungal population, but it was anyhow linked to selection of the resistant strain and subsequent treatment failure [182,183]. As for tolerance, fungal isolates may show different grades of heteroresistance as hypothesized through a mouse model of *C. glabrata* kidney infection, where highly heteroresistant isolates correlated with higher percentages of persistent infections [182]. Higher levels of heteroresistance could be associated with clinically relevant consequences in humans. The proposed underlying genetically based resistance mechanism for these two fungal pathogens is target drug/efflux pump gene aneuploidy, but still no consensus among researchers has been reached as aneuploidy could only partially explain the resistant phenotype [183,184]. Another clinically relevant issue related to heteroresistance is that it cannot be detected at standard antimicrobial susceptibility tests, this is caused by the reduced number of microorganisms constituting the heteroresistant population [185,186].

In the case of *C. parapsilosis sensu stricto,* heteroresistance to echinocandins was appointed by Zhai and colleagues to be correlated with prophylaxis failure, thus enhancing the risk of breakthrough infections [185]. Rates of *C. parapsilosis* echinocandin heteroresistance ranged between 0.1% and 0.01%, within an otherwise fully susceptible colony [92,185].

## 3. Epidemiological Landscape of *Candida parapsilosis sensu stricto* Resistance

Among all antifungal drugs, azoles are the most studied in terms of antimicrobial resistance for *C. parapsilosis* with rates higher than those reported for all other drugs and continuously increasing in the last twenty years [17,40,92]. According to the 2006–2016 SENTRY surveillance study, 3.9% of *C. parapsilosis* isolates analyzed were resistant to fluconazole [40], and differences were observed according to the geographical area, with 4.6% of *C. parapsilosis* strains isolated from European countries and 4.3% from Latin America. In a meta-analysis from Yamin and colleagues, pooled prevalence of fluconazole resistance was 15.2% up to 2022 [153]. Voriconazole resistance rates from the same investigation reported a pooled prevalence of 4.7% in the meta-analysis from Yamin [153] and high cross-resistance rates with fluconazole (32.7% of fluconazole-resistant isolates susceptible to voriconazole) into the SENTRY report [40]. However, further data from monocentric studies revealed higher rates in fluconazole-resistance than the average reported, especially from Europe (10–20% Spain and Greece; 20–30% Italy; 30–40% Turkey) Latin America (10–20%) and South Africa (40–60%) [2,3,6,7,8,92,187,188,189,190,191,192]. Among all, Govender et al. in 2016 described an astonishing rate of fluconazole-resistance in the South African province of Guateng, with only 37% of fluconazole-susceptible *C. parapsilosis* isolated from bloodstream infections [8].

Data regarding mutations found in azole-resistant *C. parapsilosis* isolates pointed out that the most frequent alterations leading to the acquisition of a resistant phenotype were the Y132F substitution in *ERG11* along with the upregulation of *MDR1* especially in European surveys and reports [92,112,187,188,193]. Considering only amino acid substitutions in the *ERG11* gene, Ceballos-Garzon and colleagues reported that Y132F was the single point mutation related to azole-resistance in Italy, South Africa, Brazil, Mexico and France [187]. Association between Y132F and K143R was observed in USA, India, Colombia, Spain and Turkey; in the last two states, in addition to Y132F and K143R also the substitution G458S was reported to be present in the same strain [187,193,194].

It has been observed that the spreading of azole-resistant *C. parapsilosis* happens through hospital outbreaks of invasive infections, especially in the case of strains harboring the Y132F substitution in the *ERG11* gene [92,188,192,195]. Along with this, azole-resistant *C. parapsilosis* is able to persist in the hospital environment causing infections even in patients without a previous history of azole exposure [92]. In-hospital transmission is carried out through contamination of health care environment, medical devices and health-care operators’ hands [196,197]. Noteworthy, most bacterial isolates harboring resistance genes, like in the case of plasmid-based carbapenemases, are selected through antibiotic pressure. In fact, its removal would restore colonization of the susceptible strain within the microbial niche. However, such a behavior is not observed in azole-resistant *C. parapsilosis* as the majority of patients affected by invasive infections during a hospital outbreak is drug naïve. Therefore, some authors suggested that fitness cost in azole-resistant *C. parapsilosis* could be equal to the susceptible strain. In addition, in-host survival time is longer for azole-resistant than -susceptible strains, thus highlighting once again that the resistant yeast pathogen is able to better adapt to host’s conditions than susceptible counterparts [4,92,188].

Among other antifungal drugs, echinocandins have been extensively used in the past few years to treat invasive infections caused by azole-resistant *C. parapsilosis* and are now considered the drug of choice in such clinical scenario [198]. However, as stated previously, this class of molecules displays fungistatic effect on *C. parapsilosis* as MIC values for echinocandin drugs are higher than for other species due the P660A polymorphism in Fks1p [104]. Despite echinocandin resistance being a seldom clinical phenomenon and rarely reported, some authors described an increased tolerance and acquired resistance [107,108], as described in studies from China [108], Turkey [106], Spain [105], Greece [140] and Brazil [100]. One study from Meletiadis et al. reported a prevalence of echinocandin-resistance in *C. parapsilosis* of 3.2% [199]. To this point, it is important to mention another study from a multicenter investigation in Spain conducted by Cantón and colleagues reporting a very low prevalence in echinocandin resistance of 0.6% in *C. parapsilosis* isolates recovered from blood cultures [200].

Data on the use of other antifungal drugs in the context of azole-resistant *C. parapsilosis* invasive infections are scarce and not often reported [201]. Despite the availability of liposomal formulation of AMB that reduced rates of adverse events, this compound is a therapeutic option reserved for only selected cases. In a clinical survey of more than 2000 isolates recovered from blood cultures, rates of AMB resistance was set up to 3% [202], according to a meta-analysis from Yamin and colleagues pooled prevalence of AMB resistance was 1.3%, with few discrepancies between different geographical regions [153]. However, considering its reduced clinical use in the context of azole-resistant *C. parapsilosis*, further data and surveys are required.

Last, flucytosine-resistance has been rarely investigated as this molecule retains a narrow clinical niche in which its use might be recommended. However, Ostrosky-Zeichner et al. and Quindos et al. reported rates of flucytosine-resistant *C. parapsilosis* in between 2–6.4%, respectively [202,203].

## 4. Conclusions

Azole-resistant *C. parapsilosis* is a major threat in public health. All three major strategies to develop resistance found in *C. albicans* have been elucidated in this species, however, the most frequently reported in clinical practice is an association of target mutations due to Y132F substitution in *ERG11* along with upregulation of *MDR1* conferring fluconazole and voriconazole cross-resistance. Data on development of resistance to other molecules, especially echinocandins, are emerging at a worrisome rate. However, the future focus of research should aim at investigating predisposing conditions and risk factors for the development of acquired resistance before the manifestation of the resistant phenotype itself. Future routinely performed microbiologic in vitro diagnostic tests should, therefore, be able to explore and report different levels of antifungal tolerance and heteroresistance in order to identify patients infected and or colonized with strains at risk of developing resistance.

## Figures and Tables

**Figure 1 jof-09-00798-f001:**
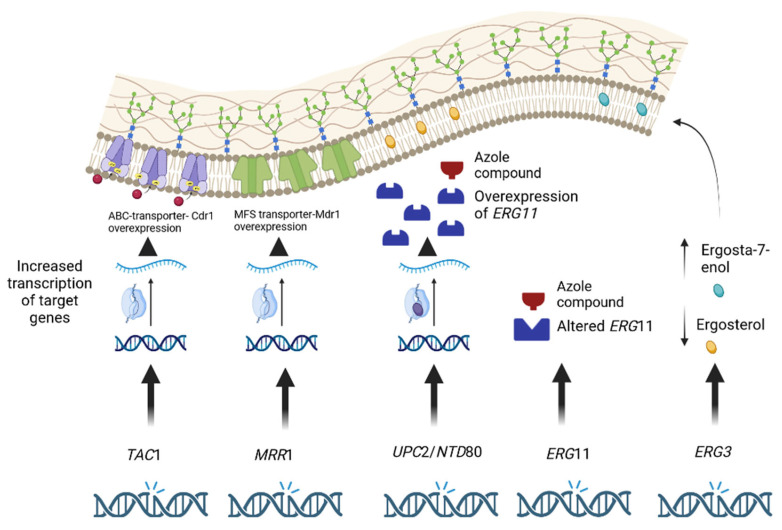
Overview of azole-resistance mechanisms in *C. parapsilosis sensu stricto.*

**Table 1 jof-09-00798-t001:** Azole and echinocandin tolerance and resistance mechanisms were chronologically reported.

Azole-Resistant *C. parapsilosis sensu stricto*				
Mechanism of Resistance	Amino Acid Change	Gene	Effect on Antifungal Drugs	Reference
Gain-of-function mutation	G583R	*MRR1*	FLU-R, VOR-R	[93,94]
Gain-of-function mutation	K873N	*MRR1*	FLU-R, VOR-R	[93,94]
Upregulation	-	*UPC2*	FLU-R, VOR-R, POS-R	[93] ^¥^
Upregulation	-	*NTD80*	FLU-R, VOR-R, POS-R	[93] ^¥^
Target change	Y132F	*ERG11*	FLU-R, VOR-R	[95]
Upregulation	L986P	*MRR1*	FLU-R, VOR-S/I	[96]
Upregulation	G650E	*TAC1*	FLU-R, VOR-R	[97]
Upregulation	L978W	*TAC1*	FLU-R, VOR-R	[97]
Loss of function	R135I	*ERG3*	FLU-R, VOR-R, POS-R	[98]
Loss of function	G111R	*ERG3*	FLU-R, VOR-R, POS-R	[99]
Upregulation	P45H	*UPC2*	FLU-R, VOR-S/I	[4]
Upregulation	Q371H	*UPC2*	FLU-R, VOR-I	[4]
Target change	K143R	*ERG11*	FLU-R	[100]
Upregulation	L518F	*TAC1*	FLU-R, VOR-R	[100]
Target change	G458S	*ERG11*	FLU-R, VOR-R	[101]
Upregulation	A854V	*MRR1*	FLU-R	[102]
Upregulation	R479K	*MRR1*	FLU-R	[102]
Upregulation	I283R	*MRR1*	FLU-R	[102]
Gain-of-function mutation	G604R	*MRR1*	FLU-R, VOR-R	[103]
**Echinocandin-Tolerant/Resistant *C. parapsilosis sensu stricto***				
**Mechanism of Resistance**	**Aminoacid Change**	**Gene**	**Effect on Antifungal Drugs**	**Reference**
Target change *	P660A	HS1-*FKS1*	ANF, CS, MYC reduced susceptibility	[104]
Target change	V595I	non-HS1-*FKS1* °	ECT ^§^, CS-I	[105]
Target change	F1386S	non-HS2-*FKS1* ^#^	ECT ^§^, ANF-R, MYC-I	[105]
Loss of function	G111R	*ERG3*	ANF-I, MYC-I/R	[99]
Target change	R658G	HS1-*FKS1*	MYC-R	[106]
Target change	E1393G	non-HS2-*FKS1* ^#^	ECT ^§^	[100]
Target change	A1422G	non-HS2-*FKS1* ^#^	ECT ^§^	[107]
Target change	M1328I	non-HS2-*FKS1* ^#^	ECT ^§^	[107]
Target change	S745L	non-HS1-*FKS1* °	ECT ^§^	[107]
Target change	S656P	HS1-*FKS1*	ANF-R, MYC-R, CS-R	[108]

^¥^ in this study only gene expression levels were evaluated, no mutation was reported. * constitutively present; ^§^ ECT = echinocandin tolerance; ° outside the Hot-spot region 1 of *FKS1*; # outside the Hot-spot region 2 of *FKS1*; FLU = fluconazole; VOR = voriconazole; POS = posaconazole; MYC = micafungin; ANF = anidulafungin; CS = caspofungin.

## Data Availability

No new data were created.

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
