# Peer review of "Candida parapsilosis sensu stricto Antifungal Resistance Mechanisms and Associated Epidemiology"

_jof, 2023, doi:10.3390/jof9080798_

Round 1

Reviewer 1 Report

The authors presented an all-embracing review of the resistance mechanisms in Candida parapsilosis. The references are up-to-date and sufficient.

Author Response

We thank the Reviewer for her/his kind appreciation.

Reviewer 2 Report

General comments

The MS is interesting and useful for the readers of Journal of Fungi. The Authors should be mention that the MS discuss mainly the C. parapsilosis sensu stricto. However, when appropriate the genetically strongly related C. orthopsilosis or C. metapsilosis should be mention as well (MIC values, resistance mechanism, etc).

Specific comments

Page 2. Lines 46-63. The sentences sometimes were complicated.

Page 2. Lines 68-71 and 75-79. For clinical resistance and persistent isolation of the pathogen from the patients are frequently originated due to the ability of the pathogen producing biofilm and the low drug concentration in the infected body sites.

Page 9. It would be interesting data from isavuconazole regarding the azole resistance.

Page 8. Rezafungin was approved as well to treat severe fungal infections some month ago, thus data from rezafungin may improve the quality of the MS.

Page 9. Lines 353-369. Though this in vitro phenomenon is interesting, its clinical relevance is questionable.

Round 2

Reviewer 2 Report

The changes looks good.